# The Zinc-BED Transcription Factor Bedwarfed Promotes Proportional Dendritic Growth and Branching through Transcriptional and Translational Regulation in *Drosophila*

**DOI:** 10.3390/ijms24076344

**Published:** 2023-03-28

**Authors:** Shatabdi Bhattacharjee, Eswar Prasad R. Iyer, Srividya Chandramouli Iyer, Sumit Nanda, Myurajan Rubaharan, Giorgio A. Ascoli, Daniel N. Cox

**Affiliations:** 1Neuroscience Institute, Georgia State University, Atlanta, GA 30302, USA; 2Center for Neural Informatics, Structures, and Plasticity, Krasnow Institute for Advanced Study, George Mason University, Fairfax, VA 22030, USA

**Keywords:** dendrite development, transcription factor, cytoskeletal regulation, translational regulation, ribosomal proteins, *Drosophila*

## Abstract

Dendrites are the primary points of sensory or synaptic input to a neuron and play an essential role in synaptic integration and neural function. Despite the functional importance of dendrites, relatively less is known about the underlying mechanisms regulating cell type-specific dendritic patterning. Herein, we have dissected the functional roles of a previously uncharacterized gene, *CG3995*, in cell type-specific dendritic development in *Drosophila melanogaster*. *CG3995,* which we have named *bedwarfed* (*bdwf*), encodes a zinc-finger BED-type protein that is required for proportional growth and branching of dendritic arbors. It also exhibits nucleocytoplasmic expression and functions in both transcriptional and translational cellular pathways. At the transcriptional level, we demonstrate a reciprocal regulatory relationship between Bdwf and the homeodomain transcription factor (TF) Cut. We show that Cut positively regulates Bdwf expression and that Bdwf acts as a downstream effector of Cut-mediated dendritic development, whereas overexpression of Bdwf negatively regulates Cut expression in multidendritic sensory neurons. Proteomic analyses revealed that Bdwf interacts with ribosomal proteins and disruption of these proteins resulted in phenotypically similar dendritic hypotrophy defects as observed in *bdwf* mutant neurons. We further demonstrate that Bdwf and its ribosomal protein interactors are required for normal microtubule and F-actin cytoskeletal architecture. Finally, our findings reveal that Bdwf is required to promote protein translation and ribosome trafficking along the dendritic arbor. These findings shed light on the complex, combinatorial, and multi-functional roles of transcription factors (TFs) in directing the diversification of cell type-specific dendritic development.

## 1. Introduction

Neurons are highly complex, polarized cells that come in an incredible variety of shapes and sizes, thanks in large part to their elaborate, cell type-specific dendritic arborization patterns that are adjusted to cover their receptive fields [1,2]. Since dendrites are primarily specialized to receive/process neuronal inputs, dendrite morphology can influence neuronal function, signal integration, and circuit assembly. Thus, understanding the biological mechanisms regulating the growth and development of dendritic arbors is of particular significance for the organization and function of the nervous system. Numerous studies have demonstrated that the acquisition of class-specific dendritic architectures is subject to regulation by complex genetic and molecular programs involving intrinsic factors and extrinsic cues including cytoskeletal regulation, transcriptional regulation, cell-signaling, and cell-cell interactions [3,4].

Transcription factors (TFs) are one class of molecules that have emerged as critical regulators of dendritic development. Different or combined TF activities have been shown to drive dendrite morphogenesis and cell type-specific dendritic diversity depending on the type of cell they are in [5,6,7,8,9,10]. In the mammalian brain, studies have also shown that the morphological identity of neurons in the cerebral cortex may be defined by the temporal or layer-specific expression of TFs [11,12]. While several studies have elaborated on the importance of TFs in regulating cell type-specific dendrite development, the complex nature of the cellular and molecular mechanisms underlying this essential biological process is only beginning to be understood [7,9,13,14,15].

*Drosophila melanogaster* has proven to be an exceptionally powerful system for identifying and characterizing transcriptional targets and cellular pathways that operate as downstream effectors of TF-mediated dendrite morphogenesis. In particular, *Drosophila* multidendritic (md) sensory neurons have emerged as a powerful model for dissecting the molecular mechanisms underlying dendrite arbor specification and diversification. Studies using this model system have revealed numerous insights into the complex intrinsic and extrinsic regulatory mechanisms underlying cell type-specific dendrite development, including dendritic outgrowth, branching, maintenance, scaling, and tiling [3,5,16]. These md neurons are categorized into four distinct morphological classes (Class I–IV; CI–CIV) based on their increasing orders of dendritic complexity [16], thereby facilitating analyses of TF-mediated programs underlying dendritic morphological diversity.

*Drosophila* research has begun to decode molecular mechanisms by which cell type-specific morphology TF networks converge on the cytoskeleton and other cellular pathways to drive dendrite arborization and spatial/functional compartmentalization [14,15,17,18,19,20,21,22,23,24,25,26,27,28,29], as well as initiator selector TF regulatory networks, which regulate the expression of other TFs, to govern a cascade of gene expression programs that drive neuronal differentiation [25,30]. Despite these significant progresses, much remains to be discovered concerning feed-forward and reciprocal TF regulatory cascades, as well as the molecular mechanisms and cell biological processes by which TFs direct dendrite development through spatiotemporal modulation of cytoskeletal dynamics and other diverse signaling pathways. Moreover, the view that different TFs are dedicated to distinct phases of neuronal morphogenesis is likely an oversimplification, and recent studies have shown that TFs continue to play important regulatory roles in postmitotic neurons for mediating distinct aspects of neuronal and dendritic development [10].

To identify and characterize morphology TFs that are required in specifying cell type-specific dendritic arborization, we previously reported on functional genomic analyses of two morphologically distinct md neuron subtypes, CI and CIV md neurons [31]. Cell type-specific CI vs. CIV transcriptomic profiling combined with a systematic loss-of-function genetic screen identified differentially enriched or depleted genes in CI vs. CIV md neurons. In this study, we characterized a limited subset of 37 TFs that were differentially expressed [31]. In addition to differentially expressed TFs, we also conducted a separate genetic screen of TFs that were expressed in these neurons, albeit not differentially expressed, from which we identified a previously uncharacterized C2H2 zinc finger transcription factor encoded by the *Drosophila melanogaster CG3995* gene. Based on our pilot screen, *CG3995* was selected for further analyses due to the highly penetrant defects in cell type-specific md neuron dendritic development. *CG3995* encodes an evolutionarily conserved zinc-finger BED-type (ZBED) protein containing a DNA binding domain found in chromatin boundary element binding proteins and transposases [32]. Members of the ZBED protein family in *Drosophila* include the proteins boundary element-associated factor of 32 kD (BEAF-32) and DNA replication-related element factor (Dref), which have been shown to antagonize each other in competing for binding to insulator or chromatin boundary elements [32]. Dysregulation of *CG3995* in md neurons leads to dendritic hypotrophy. Mutant neurons exhibit a “dwarfed” phenotype due to the proportional reductions in both dendritic branching and overall dendrite growth, and we propose that this gene be named *bedwarfed* (*bdwf*) owing to the zinc-BED domain and the dendritic growth/branching defect leading to a stunted arborization pattern. Phenotypic analyses indicate Bdwf primarily regulates dendrite growth and branching via at least two distinct mechanisms in md neurons involving transcriptional and translational regulation. At the transcriptional level, we demonstrate that the Cut homeodomain transcription factor, which is known to exhibit cell type-specific expression and drive cell type-specific dendritic arborization in md neurons [33], positively regulates Bdwf expression, where this protein acts as a downstream effector of Cut-mediated dendrite morphogenesis. Conversely, Bdwf overexpression negatively regulates Cut expression, revealing a reciprocal regulatory relationship between these TFs. At the translational level, proteomic analyses reveal that Bdwf primarily interacts with ribosomal proteins in CIV md neurons to regulate dendritic and cytoskeletal architecture. Furthermore, our data indicate that Bdwf is required for the proper trafficking of ribosomes along the dendritic arbor in both CI and CIV md neurons. Finally, we demonstrate that *bdwf* knockdown inhibits protein translation suggesting that Bdwf regulates dendritogenesis by promoting global protein synthesis, perhaps through Bdwf interactions with ribosomal proteins. Collectively, these results provide new insights into the complex combinatorial and multi-functional roles of TFs in determining cell type-specific dendrite development.

## 2. Results

### 2.1. Bdwf Dysregulation Results in Dendritic Hypotrophy

The *bdwf* gene was identified in a RNAi-mediated phenotypic screen designed to identify predicted TF/DNA binding proteins that exhibit functional roles in regulating cell type-specific dendrite morphogenesis. The *bdwf* gene encodes a single mRNA isoform bearing two coding exons and is predicted to produce a protein of 322 amino acids (37.5 kDa) (Figure 1A). Bdwf belongs to the C2H2 family of zinc-finger, BED-type (ZBED) transcription factors and bears N-terminal ZBED (amino acids 6–47) and Myb/SANT (amino acids 78–143) DNA binding/protein-protein interaction domains. To dissect the functional role(s) of *bdwf* in regulating cell-type specific dendritogenesis, we performed loss-of-function (LOF) and gain-of-function (GOF) analyses via gene-specific RNAi knockdown, mutant analysis, or overexpression of full-length *bdwf* in multiple md neuron subclasses (CI, CIII, and CIV). For mutation analysis, we identified a P{XP} transposon insertion in the *bdwf* 5′ region, *bdwf^d05488^* (Figure 1A). Homozygous mutation for *bdwf^d05488^* displayed dendritic hypotrophy, including highly collapsed terminal branching and reduced field coverage in CIV md neurons (Appendix A). To verify the specificity of the *bdwf^d05488^* allele, we performed rescue experiments. Our results revealed that the observed dendritic defects were fully rescued when a single copy of *UAS-bdwf-FLAG-HA* was expressed in CIV neurons in a homozygous *bdwf^d05488^* mutant background (Appendix A). This reveals a cell-autonomous requirement for *bdwf* in these neurons to promote dendritic arborization. To assess the putative role of *bdwf* in regulating cell type-specific dendritogenesis, we used the *GAL4-UAS* binary expression system for RNAi-mediated knockdown of *bdwf* in CI, CIII, or CIV md neurons. Two independent, gene-specific RNAi lines (*bdwf-IR*) with no predicted off-targets were tested and the transgenic line displaying stronger phenotypic penetrance was used for all subsequent RNAi analyses. CIV-specific *bdwf-IR* knockdown was highly penetrant and resulted in a significant dendritic hypotrophy manifesting as reductions in both dendritic branching and length in all three CIV md neuron subtypes (ddaC, v’ada, and vdaB) (Figure 1B,C,K,L). Across the three CIV md neuron subtypes, *bdwf-IR* reduced total dendritic length (TDL) and total dendritic branches (TDBs) in CIV neurons by an average of −26% and −27.7%, respectively (Figure 1B,C,Q). *bdwf* LOF phenotypes were most pronounced in CIII neurons (ddaF). In contrast to the elaborate space-filling branching pattern of CIV neurons, CIII neurons are characterized by the presence of filopodial-like terminal branches (Figure 1B,E). When *bdwf-IR* was expressed under the control of a *CIII-GAL4* driver, we observed an average −48.5% reduction in TDL and −53% reduction in TDB (Figure 1E,F,M,N,Q). *bdwf-IR* knockdown in CI neuron subtypes (ddaE, vpda) resulted in a similar degree of dendritic hypotrophy as observed in CIV neurons, leading to an average reduction of −29.6% and −24.7% in TDB and TDL, respectively (Figure 1H,I,O–Q).

Since *bdwf* LOF resulted in dendritic hypotrophy, we next sought to investigate how *bdwf* overexpression (*bdwf-OE*) may impact dendritic morphogenesis using the same *UAS-bdwf-FLAG-HA* transgene used in rescue studies described above. *bdwf*-OE phenotypic analyses were conducted in CI, CIII, and CIV md neurons, resulting in mild-to-severe dendritic hypotrophy as quantified by reduced TDL and/or TDB. CIII neurons displayed the strongest phenotypic defects with an average −38.2% reduction in TDB and −24.1% reduction in TDL (Figure 1G,M,N,Q). On average, CI md neurons displayed a moderate reduction in TDL and TDB of −14.9% and −12.6%, respectively (Figure 1H,O–Q). Overexpression of *bdwf* had the mildest effect on CIV neurons, resulting in an average −14.3% reduction in TDB, albeit this defect was only observed in the dorsal most CIV md neuron subtype (ddaC), but showed no significant change (−0.1% reduction) in TDL relative to control (Figure 1D,K,L,P). Collectively, LOF and GOF phenotypic analyses indicate that cell type-specific dendritic morphogenesis is sensitive to the Bdwf levels in CI, CIII, and CIV md neurons.

### 2.2. Bdwf Regulates Proportional Dendritic Growth and Branching in Morphologically Divergent Neurons

Qualitative analyses of *bdwf* LOF/GOF phenotypes revealed that *bdwf* LOF mutants appear to have “dwarfed” dendritic arbor morphology relative to control. This manifested as concomitant reductions in total dendritic growth and higher-order branching across md neuron subclasses, whereas GOF analyses suggested that there were larger effects on higher-order dendritic branching, most notably in the more morphologically complex CIII and CIV neurons (Figure 2). To quantify the subclass-specific effects of *bdwf* dysregulation on cell type-specific arborization patterns, we calculated dendritic branch density (DBD) by normalizing the TDB by TDL for each md neuron subtype. This metric is effectively the inverse of the average branch length, which is useful to distinguish between arbor size and complexity [34]. In five of the six neuronal subtypes we analyzed, *bdwf-IR* knockdown displayed no statistically significant changes in branch density compared to controls (ddaC, v’ada, ddaF, ddaE, vpda). The CIV vdaB ventral subtype was the only exception, showing a mild DBD reduction (−9.5%) (Figure 2A,B,D–G). The observation that *bdwf* LOF mutants quantitatively resemble their WT control counterparts concerning DBD indicates that the reduction in dendritic branching is highly proportional to the reductions in total dendritic length (Figure 2G). These results suggest that Bdwf plays a role in regulating proper dendritic arborization by facilitating proportional dendritic growth and branching in morphologically divergent neurons. Conversely, quantitative analyses of *bdwf* GOF phenotypes revealed significant reductions in DBD for CIII and CIV neurons, whereas no significant change was observed in CI neurons (Figure 2A,C–F). CIII neurons displayed the most reduction in branch density (−31.2%), followed by CIV neurons (−14.25%), whereas CI neurons displayed no significant change in DBD (+2.5%) relative to control (Figure 2G). This suggests that Bdwf overexpression exerts distinct effects on dendritic branching vs. growth in more morphologically complex CIII and CIV neurons as compared to simpler CI neurons, which may be due to cell type-specific differences in Bdwf interactors across md neuron subclasses.

**Figure 2 ijms-24-06344-f002:**
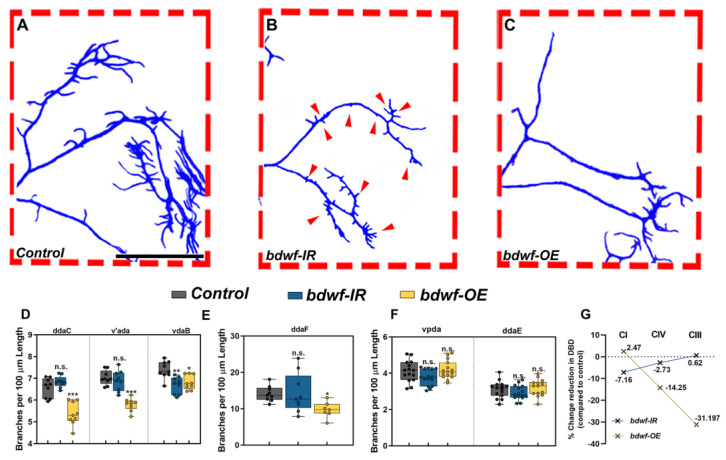
***bdwf is required for proportional dendritic growth and branching.*** (**A**–**C**) Compared to wild-type (**A**), Bdwf LOF (**B**) results in a proportional reduction of TDB with respect to TDL, as seen by the reduction in the size and number of CIII dendritic termini (red arrows). In contrast, Bdwf GOF (**C**) results in the suppression of dendritic termini. Scale bar, 100 µm. (**D**–**F**), Quantification of TDBD for CIV, III, and I neurons (**D**–**F**), respectively. (**G**) Class-specific effects of Bdwf LOF and GOF on TDBD represented as percentage change from control. * *p* ≤ 0.05, ** *p* ≤ 0.01, *** *p* ≤ 0.001, n.s. = not significant (One-way ANOVA with Dunnett’s multiple comparison test). *n* = 14–15 for CI, *n* = 7,9 for CIII and *n* = 8,9 for CIV. Genotypes: *Gal4^ppk^, GAL80^ppk^, UAS*–*mCD8::GFP/+; +/+* or *GAL4^ppk^,GAL80^ppk^,UAS*–*mCD8::GFP/+;UAS*–*bdwf*–*IR/+* or *GAL4^ppk^,GAL80^ppk^,UAS*–*mCD8::GFP/+;UAS*–*bdwf*–*FLAG*–*HA/+* (**A**–**C**,**E**,**G**). *GAL4^477^, UAS*–*mCD8::GFP/+; +/+* or *Gal4^477^, UAS*–*mCD8::GFP/+; UAS*–*bdwf*–*IR/+* or *Gal4^477^, UAS*–*mCD8::GFP/+; UAS*–*bdwf*–*FLAG*–*HA/+* (**D**,**G**). *GAL4^221^,UAS*–*mCD8::GFP/+ or GAL4^221^,UAS*–*mCD8::GFP/UAS*–*bdwf*–*IR* or *GAL4^221^,UAS*–*mCD8::GFP/UAS*–*bdwf*–*FLAG*–*HA* (**F**,**G**).

### 2.3. Bdwf Exhibits Nucleocytoplasmic Localization in Md Neurons

Given the LOF and GOF phenotypes associated with *bdwf* dysregulation, we sought to investigate the expression and subcellular localization of Bdwf protein in md neuron subclasses. Bdwf protein expression was discovered in multiple md neuron subclasses using immunohistochemistry on third instar larval filets. Bdwf exhibited punctate nucleocytoplasmic localization with qualitatively higher levels of cytoplasmic expression in control md neurons (Figure 3A,B). In addition to Bdwf expression in md neuron subclasses, we also observed expression in adjacent epithelial and/or muscle cells, between which the md neurons are sandwiched (Figure 3A,B). To verify the specificity of the antibody and to document the efficacy of the *bdwf-IR* knockdown transgene, we analyzed Bdwf expression levels in control vs. *bdwf-IR* CIV md neurons, which revealed a significant reduction of Bdwf protein expression levels (Appendix A). In contrast to the more cytoplasmically distributed Bdwf puncta observed in control neurons, *bdwf* overexpression using a pan-md *GAL4* driver led to a dramatic shift to a predominantly nuclear subcellular localization for Bdwf expression (Figure 3C,D). To dissect the putative effects of Bdwf domains on subcellular localization, we generated structure-function mutant transgenes that express either the BED DNA binding domain alone (*UAS-bdwf-BED-HA*) or a version of *bdwf* lacking only the BED domain (*UAS-bdwf-ΔBED-myc*). Using a CIV-specific *GAL4* driver, we expressed these structure-function *bdwf* variants and immunostained for epitope tags associated with each transgene to visualize subcellular distribution. Expression studies revealed that the transgene lacking the Bdwf BED domain (*UAS-bdwf-ΔBED-myc*) was localized to the cytoplasm (Figure 3E,F), whereas the transgene bearing only the Bdwf BED (*UAS-bdwf-BED-HA*) was predominantly localized to the nucleus (Figure 3G,H). These data suggest that the Bdwf BED domain regulates nuclear localization.

### 2.4. Cut and Bdwf Exhibit a Reciprocal Regulatory Relationship with Respect to Expression

Previous studies have demonstrated that TF-mediated regulation of cell type-specific md neuron dendritic diversification is subject to TF regulatory networks whereby select TFs regulate the expression of other TFs to target downstream effectors that drive morphological diversity [10,25,30]. As LOF and GOF of analyses of *bdwf* revealed the strongest phenotypic defects in CIII md neurons, we hypothesized that Bdwf may operate in a TF regulatory network with the Cut (Ct) homeodomain transcription factor, which is expressed at the highest levels in CIII neurons and is known to drive cell type-specific dendritic arborization in select md neuron subtypes (Classes II-IV) [33]. Moreover, there is a high degree of phenotypic similarity with respect to dendrite morphogenesis defects observed for *bdwf* and *ct* dysregulation. For example, previous studies have demonstrated that the formation of the actin-rich terminal filopodial branches, characteristic of CIII neurons, is dependent upon Ct [33], which is found to be significantly reduced in *bdwf* GOF mutants (Figure 2E).

To explore a putative regulatory relationship between Ct and Bdwf, we first tested a hypothesis that Ct may positively regulate *bdwf* expression. To address this hypothesis, we isolated control CI md neurons and CI neurons in which we ectopically overexpressed Ct and then performed qRT-PCR analyses to measure potential effects on *bdwf* mRNA expression levels. We observed that expression of a single copy of *ct* (*UAS-ct*) in CI neurons led to a four-fold increase in *bdwf* mRNA levels (Figure 4A). In comparison, *ct* mRNA expression levels were found to be upregulated by ~14-fold for control CI neurons (Figure 4A). To assess how changes in mRNA levels may translate to protein levels, we ectopically overexpressed Ct in CI neurons and assayed for changes in native Bdwf protein levels via immunostaining. Ectopic Ct expression in CI neurons led to over 110% increase in Bdwf protein expression in these neurons relative to controls (Figure 4B,D–K). In addition, we noted a mild but significant increase in nuclear Bdwf levels when Ct was ectopically expressed, as quantified by the nuclear to whole soma ratio of Bdwf expression (Figure 4C). Expression analyses revealed Bdwf labeling in CI md neurons (Figure 3A and Figure 4D–G). This suggests that, at least in CI neurons, Ct is not required for Bdwf protein expression, given that Ct is not normally expressed at detectable levels in these neurons [26,33]. To examine whether this holds true for neurons that normally express Ct, we used mosaic analysis with a repressible cell marker (MARCM) to generate *ct^c145^* null mutant CIV neuron clones and stained them for Bdwf. These analyses revealed that *ct* is not solely required for Bdwf protein expression in Ct-positive CIV neurons (Appendix A). Collectively, these analyses indicate that Cut has a positive effect on the expression of *bdwf* mRNA and Bdwf protein expression in md neurons.

Given the positive feed-forward regulatory relationship we observed between Ct and Bdwf, we next sought to investigate whether Bdwf may exert any reciprocal regulatory effects on Ct expression. Since Bdwf overexpression induced a reduction in dendritic branch density in Cut positive neurons (CIII and CIV neurons), we hypothesized that Bdwf and Cut may exhibit a reciprocal regulatory relationship. To test this, we overexpressed Bdwf in CIV neurons and quantified Cut immunostaining levels normalized to adjacent wild-type CIII neurons. Expression of a single copy of *ct-IR* in CIV neurons resulted in a strong suppression of Cut immunostaining levels by ~−40% (Figure 4R). Bdwf overexpression in CIV neurons likewise resulted in strong suppression of Cut expression, to nearly the same levels as observed with *ct-IR* expression (−36.1%, Figure 4L–R). This indicates that Bdwf can negatively regulate Cut in a reciprocal feedback loop when overexpressed in these neurons. To determine whether Bdwf may be required for Ct expression, we specifically knocked down *bdwf* in CIV neurons, which resulted in a 19% increase in Ct immunostaining when compared to controls. However, this change, while trending (*p* = 0.057), did not rise to the level of statistical significance (Figure 4R). Taken together, these data suggest that the Bdwf overexpression-induced dendritic hypotrophy phenotype is likely due, at least in part, to the negative feedback effects on Ct expression.

### 2.5. Bdwf Functions as a Downstream Effector of Cut-Mediated Dendritic Arborization

Cut has previously been shown to induce dramatic dendritic hypertrophy in a dose-dependent manner in CI neurons [33]. We investigated whether Cut-induced dendritic complexity occurs via a Bdwf-dependent pathway to investigate the functional consequence of the Cut-Bdwf regulatory relationship. We hypothesized that if Bdwf functions as a downstream effector of Cut-mediated dendritic development, then disrupting *bdwf* in a genetic background sensitized by ectopic Ct overexpression should lead to a suppression of Cut-mediated dendritic hypertrophy. To test this hypothesis, we generated a line that stably expressed *UAS-cut* under the control of the CI md neuron driver *GAL4^221^*, which was outcrossed to either *UAS-bdwf-IR*, *UAS-bdwf*, *UAS-cut*, or *cut-IR* to assay the respective phenotypic effects on CI dendrite morphogenesis. To control for *GAL4-UAS* titration, we expressed *UAS-mCD8::RFP* in the Cut overexpression control background. As expected, ectopic expression of Cut in CI neurons results in dramatic dendritic hypertrophy, which manifests as a sharp increase in lower-order dendritic branch extension as well as the *de novo* formation of numerous terminal dendritic filopodia-like protrusions, relative to wild-type controls (Figure 5A,B,E,F) [33]. Expression of *UAS-cut* (+ *UAS-mCD8::RFP*) in CI vpda neurons resulted in an over 7.5-fold increase in TDB and a 3-fold increase in TDL relative to wild-type controls (Figure 5D–F). We first validated the effectiveness of this system by expressing *cut-IR*, which resulted in strong suppression of dendritic hypertrophy (Figure 5C,E,F). Co-expression of *UAS-cut* and *UAS-bdwf-IR* likewise resulted in suppression of Cut-mediated dendritic hypertrophy with respect to dendritic growth and branching (Figure 5D–F). These results indicate that Bdwf is required for Cut-induced dendritic hypertrophy in CI neurons.

Given that Cut is not normally expressed in CI md neurons, we sought to test whether Bdwf overexpression could rescue dendritic hypotrophy defects observed when *ct* is knocked down in CIV md neurons, which normally express Ct protein. CIV-targeted expression of *ct* RNAi (*ct-IR*) reduces the total dendritic length and the number of dendritic branches significantly compared to controls (Appendix A). When we simultaneously knocked down *ct* (*ct-IR*) and overexpressed *bdwf* (*UAS-bdwf*) in CIV neurons, we observed a partial rescue resulting in a significant recovery of the reductions we observed for total dendritic length and total dendrite branches (Appendix A). While the rescue did not achieve a full recovery of *ct*-mediated dendritic hypotrophy back to control morphology, this was predicted as numerous genes have been previously identified as downstream effectors of *ct*-mediated dendritic development [9,17,19,21,22,23,25,26,27,29]. Collectively, these findings reveal that Bdwf functions as a downstream effector of Cut-mediated dendritic arborization.

### 2.6. Bdwf Interacts and Colocalizes with Ribosomal Proteins

Endogenous Bdwf protein expression revealed strong cytoplasmic labeling, so we sought to investigate potential mechanisms by which Bdwf may operate in the cytoplasm to regulate dendrite morphogenesis. To this end, we conducted affinity purification of Bdwf interacting proteins in combination with mass spectroscopy (MS) analyses on age-matched larval tissue. In addition to purifying Bdwf, MS analyses revealed over 70 proteins interactors of Bdwf, almost half of which were ribosomal proteins (~44%; Figure 6A; Appendix A). The gene ontology (GO) term “cytosolic ribosome” was found to be the most statistically enriched category in this dataset (~30-fold enrichment, *p* = 2.08 × 10^−42^) (Appendix A). Furthermore, co-immunostaining of Bdwf and one of the protein interactors, ribosomal protein S6 (RpS6), revealed colocalization of these proteins in the cytosol of CIV md neurons (Figure 6B). These data reveal that Bdwf physically interacts with and colocalizes with ribosomal proteins.

We reasoned that if Bdwf interactors were functionally important for dendritogenesis, disrupting members of this protein complex should phenocopy *bdwf* LOF disruptions at least partially. To functionally validate the role of Bdwf interactors in dendritogenesis, we selected genes encoding both large and small subunit ribosomal proteins that were enriched in our MS dataset and performed gene-specific RNAi analyses in CIV neurons. CIV-targeted RNAi against *RpS24, RpL4, RpL22*, and *RpL31* resulted in dendritic hypotrophy consistent with the defects observed with *bdwf* RNAi knockdown, including significant reductions in TDL and TDB relative to wild-type controls (Figure 6C–F). In addition, except for *RpS24-IR*, the reduction in the TDL and TDB for the ribosomal protein knockdowns was not significant when compared to *bdwf* RNAi (Figure 6C–H). These results suggest that Bdwf might likely be functioning in conjunction with multi-protein ribosomal complexes to promote dendritic arborization.

Cell type-specific dendritic architecture is mediated by the organization and dynamics of cytoskeletal fibers [26,29,35,36]. In the case of *Drosophila* md neurons, previous studies have demonstrated that the local distribution and organization of microtubules (MT) and F-actin fibers are sufficient for constraining arbor morphology [37]. Given that LOF for *bdwf* and ribosomal protein-encoding genes produced similar defects in dendritic morphogenesis, we next examined how the disruption of these genes may impact the underlying dendritic cytoskeleton. We investigated MT and F-actin cytoskeletal fibers in CIV neurons using multi-fluorescent reporter lines that label F-actin (*UAS-GMA::GFP*) or MTs (*UAS-mCherry::Jupiter*) [26]. The F-actin and MT intensities were then quantified by multichannel digital reconstructions as previously described [38]. CIV knockdown of *bdwf* or genes encoding ribosomal proteins severely diminished the MT signal. In the case of *bdwf-IR*, a percentage change from control analysis revealed that, at 20 µm from the soma, there was ~50% reduction in MT signal, which kept decreasing on dendrites distal to the soma with a nearly complete loss in MT signal at ~400 µm from the soma (Figure 7A’,A’’,C’,C’’,H). Apart from *RpS24*, knockdown of *RpL4, RpL22,* and *RpL31* ribosomal proteins had a similar effect on MT with nearly 70% reduction in MT signal at 20 µm from the soma, which progressively decreased distal to the soma (Figure 7E’,E’’,F’,F’’,G’,G’’,H). *RpS24* knockdown had a milder effect on MT signal, with only ~40% reduction in MT signal at 20 µm from the soma. At the farthest distance from the cell body, 540 µm, the MT signal was reduced by ~77% (Figure 7D’,D’’,H). In contrast to MTs, *bdwf-IR* led to a redistribution of the F-actin signal. Percent change from control analysis showed that at ~60–160 µm from the soma, F-actin signal increased by more than 20%; however, beyond this distance from soma, the F-actin signal continued to decrease on arbor most distal from the soma (Figure 7A,C,I). Similar to their effect on MTs, the knockdown of *RpS24, RpL4, RpL22,* and *RpL31* ribosomal proteins led to a decrease in the F-actin signal as a function of distance from the soma (Figure 7D–G,I). Previous research from our lab has demonstrated that Ct can regulate effector molecules that target the cytoskeleton, influencing dendritic morphogenesis [26,29]. Moreover, we have also shown that Ct regulates the expression of ribosomal proteins to modulate dendritic complexity [26]. Since our study establishes that Cut and Bdwf have a reciprocal relationship, we examined defects associated with cytoskeletal architecture upon *ct* knockdown in CIV neurons, which revealed severely reduced F-actin labeling throughout the dendritic arbor and progressively diminishing MT signal as a function of distance from the soma (Figure 7B,B’,B’’,H,I). Taken together, our results indicate that *ct*, *bdwf*, and genes encoding ribosomal proteins are required to support dendritic cytoskeletal organization and stability.

### 2.7. Bdwf Is Required for Ribosomal Trafficking and Protein Translation along theDendritic Arbor

A previous study in *C. elegans* has shown that disruption of MTs alters ribosome localization in axons [39]. Our cytoskeletal analyses showed that the knockdown of *bdwf* severely disrupted MT stability along the dendritic arbor. Therefore, we sought to determine the effect of *bdwf-IR* on ribosome trafficking along the dendritic arbor. We expressed GFP-tagged RpL10Ab [40] under the control of either CI or CIV *GAL4* drivers. This GFP-tagged RpL10Ab is reported to be incorporated in both polysomes and monosomes [40], so we used it as a proxy to investigate ribosome localization along the dendrites. In both CI and CIV control neurons, ribosome signal is found distributed along the arbor and at dendritic branch points (Figure 8A,A’,E,E’). *bdwf* knockdown severely impaired ribosome trafficking along the arbor (Figure 8B,B’,F,F’). Quantitative analyses revealed that in both CI and CIV neurons, there was a significant reduction in ribosome density along the dendritic arbor (Figure 8C,G) as well as in the number of branch points with ribosomes (Figure 8D,H) in *bdwf-IR* compared to control. This suggests that Bdwf is required for ribosomal trafficking on dendritic arbors in these neurons.

Our data demonstrate that Bdwf co-localizes with ribosomal proteins and is also required for ribosome localization along the dendritic arbor. We next wanted to determine the effect of *bdwf* knockdown on global protein translation. We expressed the mutant methionyl tRNA-synthetase (*UAS-dMetRS^L624G^-3xmyc*) under the control of a CIV *GAL4* in control and *bdwf-IR* conditions. MetRS^L274G^ causes methionyl-tRNA to be charged with azidonorleucine (ANL), allowing its incorporation into proteins, which can then be used to visualize translated proteins through biorthogonal click chemistry labeling [41]. In control neurons, de novo proteins tagged with the red fluorescent dye tetramethyrhodamine (TAMRA) can be seen both in the cell body and along the dendrites (Figure 8I,I’). In *bdwf-IR* animals, the TAMRA signal was restricted to the cell body with no discernable signal in the dendrites (Figure 8J,J’). Quantitative analysis showed a significant reduction in the levels of translated protein in *bdwf-IR* compared to controls (Figure 8K). Thus, our findings show that Bdwf is required for ribosome trafficking along the dendritic arbor as well as protein translation in md neurons.

## 3. Discussion

The purpose of this paper is to describe the characterization of Bdwf, a novel zinc-BED domain-containing transcription factor, and its roles in regulating cell-type specific dendritogenesis in the *Drosophila* sensory neurons. Loss of *bdwf* led to dendritic hypotrophy in all three neuronal subtypes analyzed (CI, CIII, CIV md neurons), and analyses indicate that Bdwf is required for proportional dendritic growth and branching. We show evidence that Bdwf exhibits nucleocytoplasmic localization. Additionally, Bdwf acts via at least two independent pathways to regulate dendritogenesis. In the complex CIII and CIV neurons, Bdwf and Cut exhibit a reciprocal regulatory relationship that promotes dendritic arborization in CIV neurons. Both Bdwf and Cut are required for MT and F-actin cytoskeletal organization and/or stability. Subsequent analyses revealed that Bdwf interacts with ribosomal proteins and is required for proper ribosome complex localization along the dendritic arbor, coupled to global protein synthesis. Position-Specific Iterated BLAST (PSI-BLAST) analysis of the Bdwf protein sequence revealed that Bdwf shares sequence similarities with MSANTD4 and Zbed4 in mice and ZBED4 in humans. While MSANTD4 has been linked to Huntington’s disease, and ZBED4 has been linked to schizophrenia and Phelan-McDermid syndrome in recent studies, little is known about these proteins' underlying mechanisms [42,43,44,45]. Our study may provide new and valuable entry points for further research on the role of these proteins in the nervous system's function and dysfunction.

Our results indicate that the BED domain aids in Bdwf nuclearization, as its deletion leads to a predominantly cytoplasmic signal. Bdwf could self-associate to form a multimeric complex, which could aid in its nuclearization. Studies of ZBED4/KIAA0636 revealed self-association via the hATC domain, which was essential for its nuclear accumulation and DNA binding [46]. However, sequence analyses of Bdwf did not identify the hATC domain. This means that if Bdwf relies on self-association for nuclear localization, it must occur via a dissimilar and unknown mechanism. Beyond sensory neuron expression, which is the focus of this study, we also found Bdwf to be expressed in a diverse set of tissues including central nervous system neurons, muscle, epithelia, and ovaries. Similar to our observation, the mouse homolog of Bdwf, ZBED4, has been reported to be expressed at different levels in many mouse tissues, including the brain, heart, ovary, and retina, and to have both nuclear and cytoplasmic localization in mouse and human retinal cones [47,48].

For neurons to effectively receive and propagate signals, they must cover their receptive fields and form synaptic connections. As an organism grows, the organs and tissues also increase in size. Tissues keep up with this increasing size by two mechanisms: either by increasing the number of cells while keeping the size of the individual cell constant or, as in the case of certain neurons, such as the Purkinje cells, the axons, and dendrites grow relative to the increase in body size [49]. Concerning dendritic arbor scaling, studies have demonstrated that animals grown under mild starvation conditions exhibit a decrease in arbor size that was proportional to the decrease in body size resulting in a miniaturized dendritic arbor [50]. In the case of miniaturized dendritic arbors, there are decreases in arbor size and total dendritic length but often unaltered terminal branching, resulting in a net increase in dendritic branch density. In contrast, neurons with aberrant Insulin/IGF signaling or TORC1 signaling result in hypotrophic, dwarfed, or simplified dendritic arbors that are distinguished from miniaturized arbors observed under starvation conditions in that there are proportional decreases in arbor size, terminal branching, and the total length, leading to dendritic branch densities that are not significantly different from control neurons [50]. Studies have shown that both cell-autonomous and non-cell-autonomous programs operate to regulate dendritic scaling vs. proportional growth and branching. Concerning non-cell-autonomous regulation of dendritic scaling, studies have demonstrated that scaling growth of dendrites in *Drosophila* md neurons requires the microRNA *bantam,* which acts in epithelial cells to diminish Akt kinase activity in adjacent neurons and thereby regulates dendritic morphogenesis [51]. In terms of cell-autonomous mechanisms for dendritic scaling, studies have demonstrated a role for the co-chaperone of HSP90, *CHORD*, which interacts with the TORC2 component, *Rictor*, to mediate dendritic scaling [50]. Our study reveals that *bdwf* LOF causes a proportional reduction in both dendritic length and branching such that the branch density remains unaffected. This suggests that there is a neural requirement for promoting proportional growth and branching of dendrites in multiple md neuron subtypes. 

We identified at least two pathways whereby Bdwf functions to regulate dendrite morphogenesis, including transcriptional and translational regulation. For transcriptional regulation, we characterized a regulatory relationship between the homeodomain TF Cut and Bdwf. Cut exhibits differential expression in morphologically diverse md neuron subtypes and is required to promote dendritic architecture in these neurons. The TFs Vestigial and Scalloped have been shown to limit Cut expression in the CII neurons [30], whereas the TF Lola promotes its expression in CIV neurons [25]. While the expression of Bdwf and Cut are independent of each other, our data indicate that these two proteins interact via a reciprocal feedback loop, wherein Cut acts upstream to promote Bdwf expression, while overexpression of Bdwf has a negative feedback effect thereby restricting the expression of Cut. Moreover, we demonstrate that Bdwf functions as a downstream effector of Cut-mediated dendritic morphogenesis, thereby adding to the complex molecular and cellular processes via which Cut exerts control over cell type-specific dendritic development [9,17,19,21,22,23,25,26,27,29]. Interestingly, the observation that Bdwf overexpression can negatively feedback to suppress Cut levels may provide some insight into the findings that both LOF and GOF for Cut produce similar dendritic hypotrophy phenotypes in CIV md neurons. Intriguingly, another BED domain bearing TF, Dref, has been shown to have an antagonistic regulatory relationship with Cut for controlling PCNA gene expression [52].

Concerning translational roles for Bdwf, co-immunoprecipitation analysis, combined with mass spectrometry data, indicates that Bdwf interacts with numerous ribosomal proteins of both the large and small subunits. Mutations in ribosomal protein genes have previously been associated with growth defects in animals, leading to a “Minute” phenotype [53]. Loss-of-function of ribosomal proteins generally leads to a defect in cellular growth, and knockdown of *RpL35A* has also been shown to cause cell growth and viability phenotypes when assayed in Kc_167_ and S2R^+^ cells [54]. In cultured mouse hippocampal neurons, loss of ribosomal proteins S6, S14, and L4 is associated with reduced protein synthesis that leads to a more simplified dendritic arbor [55], while loss of *RpL22* led to a disruption in dendritic morphology in CIV md neurons [56]. Mutations in *RpS24* are associated with Diamond-Blackfan Anaemia and in *Drosophila* have been shown to cause developmental delays which were rescued by expressing synaptic vesicle proteins in serotonergic neurons [57,58]. The observation that *bdwf* mutation leads to a proportional reduction in dendritic growth and branching may be explained, at least in part, by Bdwf's interaction with ribosome components, as knockdown of these genes results in severe dendritic hypotrophy. Moreover, the proper functioning of the nervous system requires neurons to respond to stimuli within minutes. Given the complexity of dendritic arbors, this is achieved through local protein translation at distal dendrites and axons [59]. The presence of ribosomes in dendrites and axons has long been established [59,60,61]. Studies have also demonstrated that activity-dependent local protein translation leads to the remodeling of the actin cytoskeleton which in turn modulates dendritic spine morphology [62]. Our data show that *bdwf* depletion disrupts ribosome trafficking along dendrites and that *bdwf* LOF leads to reduced translation in CIV neurons. Concerning the cytoskeleton, Cut has previously been shown to regulate dendritic architecture via downstream effectors that direct F-actin and MT dendritic cytoskeletal organization and stability [17,26,29]. Similarly, genetic depletion of *bdwf* or genes encoding ribosomal proteins destabilizes MT assembly and reduces F-actin rich dendritic branching, implying that Bdwf acts in concert with ribosomal proteins to promote dendritic growth by modulating both F-actin and MT cytoskeletal architecture.

We propose a model of nucleo-cytoplasmic Bdwf activity whereby the constellation of functional roles in ribosomal trafficking, protein translation, and cytoskeletal architecture, together with the TF regulatory relationship Bdwf shares with Cut, provides mechanistic insights for how this gene directs cell-type specific dendritic arborization (Figure 9). Both Cut and Bdwf have been shown to control the F-actin and MT cytoskeletons, but it remains to be seen if this is a direct effect of Bdwf in the nucleus.

## 4. Materials and Methods

### 4.1. Drosophila Strains

*Drosophila* stocks were raised on standard cornmeal-molasses-agar media at 25 °C. Fly strains used in these studies were either generated (*UAS-bdwf*, *UAS-bdwf-BED-HA*, *UAS-bdwf-ΔBED-myc*) or obtained from Bloomington or Vienna *Drosophila* stock centers. *bdwf^d05488^* line was obtained from the Exelixis collection at Harvard Medical School [63]. The following gene-specific *UAS-RNAi* (*IR*) lines were tested: *bdwf-IR* (JF02831; GD10700); *cut-IR* (JF03304); *RpS24-IR* (v104676); *RpL4-IR* (v101346); *RpL22-IR* (v104506); *RpL31-IR* (v104467). *GAL4* strains for md neuron subtypes included: *GAL4^221^, UAS-mCD8::GFP* (*CI-GAL4*); *ppkGAL4, ppkGAL80*, *UAS-mCD8::GFP* (*CIII-GAL4*); *GAL4^477^,UAS-mCD8::GFP,/CyO,tubP-GAL80* (*CIV-GAL4*); *GAL4^ppk1.9^,UAS-mCD8::GFP* (*CIV-GAL4*) *GAL4^21−7^,UAS-mCD8::GFP* (*pan-md-GAL4*). Other strains included: *UAS-cut*; *UAS-mCD8::RFP*; *w, ct^c145^, FRT^19A^/y^+^, ct^+^, Y* [33]; *y,w,tubP-GAL80,hsFLP,FRT^19A^; GAL4^109(2)80^,UAS-mCD8::GFP; UAS-RpL10Ab-GFP*; *UAS-dMetRS^L624G^-3xmyc* [41]; *UAS-Luciferase-IR; UAS-GMA::GFP;GAL4^477^, UAS-mCherry::Jupiter* [26]; *GAL4^Act5c^; UAS-CD4-tdTomato*. *Oregon-R* was used as a wild-type strain. UP-TORR (https://www.flyrnai.org/up-torr/ (accessed on 1 February 2023)) was used to computationally assess predicted off-target effects for gene-specific RNAi constructs [64], and all IR lines used have no predicted off-target effects.

### 4.2. Transgenic Strain Generation

For this study, we generated a full-length Bdwf overexpression transgene (*UAS-bdwf-FLAG-HA*) as well as structure-function variants (*UAS-bdwf-BED-HA* and *UAS-bdwf-ΔBED-myc*). To generate the full-length overexpression transgene, a full-length cDNA for the sole *bdwf* mRNA isoform (BDGP clone LD44187) was subcloned into *pUAST* with a C-terminal insertion of FLAG and HA epitope tags. We custom synthesized versions of *bdwf* (Genscript, Piscataway, NJ, USA) that express only the BED domain (amino acid 1–47) or that lack the BED domain (ΔBED; removal of amino acids 6–47) for structure-function variants. These custom gene synthesis products were then Gateway^®^ subcloned into the pTWH plasmid for the BED-only domain or into the pTWM plasmid for the ΔBED variant. pTWH is a *pUAST* vector bearing a C-terminal HA epitope tag, while pTWM is a *pUAST* vector bearing a C-terminal myc epitope tag. All three *bdwf* transgenes were generated by ΦC31-mediated integration with targeting to 2R (51C1) (Best Gene, Chino Hills, CA, USA). All constructs were sequenced and checked for precision.

### 4.3. Cell Isolation

The isolation and purification of CI da neurons were performed as previously described [65]. Briefly, 40–50 age-matched third instar larvae having the genotypes of either +/+; *GAL4^221^,UAS-mCD8::GFP* or *UAS-Cut/+; GAL4^221^,UAS-mCD8::GFP* were collected and washed several times in ddH20. The larvae were then rinsed in RNAse away and ddH20 before being dissected. The tissue was then dissociated using a combination of enzymatic and mechanical perturbations to yield single-cell suspensions, which were filtered using a 30 µm membrane. The filtrate is then incubated with superparamagnetic beads (Dynabeads MyOne Streptavidin T1, Invitrogen, Waltham, MA, USA) coupled with biotinylated mouse anti-CD8a antibody (eBioscience, San Diego, CA, USA) for 60 minutes. Finally, the CI neurons attached to the magnetic beads were then separated using a powerful magnetic field. The isolated neurons were washed at least five times to remove any potential nonspecific cells, and the quality and purity of the isolated neurons were assessed under a stereo-fluorescent microscope equipped with phase contrast for examining the number of fluorescent/nonfluorescent cells. Only if the isolated cells were free of cellular debris and non-specific (i.e., non-fluorescing) contaminants were they retained. The purified CI neurons were lysed in RNA lysis buffer and stored until needed for qRT-PCR analyses.

### 4.4. qRT–PCR Analysis

The qRT–PCR analysis of control and Cut overexpressing CI da neurons was performed in triplicates and repeated independently twice. The expression levels of *cut* and *bdwf* were assessed by qRT–PCR. The values obtained from these analyses were normalized to the endogenous control (*RpL32*). The levels relative to those observed in the flow-through fraction were calculated using the ΔΔCτ method [66]. Pre-validated Qiagen QuantiTect Primer Assays (Qiagen, Germantown, MD, USA) were used for *cut* (QT00501389), *bdwf* (QT00976549), and *RpL32* (QT00985677).

### 4.5. Phenotypic Screening, live Image Confocal Microscopy, and Morphometric Quantitation

Live imaging was performed as previously described [31]. Briefly, *GAL4^477^,UAS-mCD8::GFP*, *GAL4^ppk^*, or *GAL4^ppk^,GAL80^ppk^,UAS-mCD8::GFP*, or *GAL4^221^,UAS-mCD8::GFP* female virgins were aged for 2–3 days, crossed to *UAS-bdwf-IR* or *UAS-bdwf-FLAG-HA* transgenic males, and reared at 29 °C. *Oregon R* flies were used as controls. For Bdwf-Ct interaction in CI neurons, *UAS-ct; GAL4^221^,UAS-mCD8::GFP* female virgin flies were crossed to either *UAS-ct*, or *UAS-bdwf-IR*, or *UAS-mCD8::RFP*, and *Oregon R* was used as controls. To analyze Bdwf-Ct interaction in CIV neurons, *GAL4^477^,UAS-mCD8::GFP; ct-IR* virgin female flies were crossed to either *UAS-CD4-tdTomato* (B35837*)* or *UAS-bdwf-IR*. *GAL4^477^,UAS-mCD8::GFP* virgin female flies that were outcrossed to *UAS-CD4-tdTomato* were used as controls. Images were acquired from at least 6–10 wandering third-instar larvae. Larvae were placed on a microscope slide, immersed in 1:5 (*v*/*v*) diethyl ether to halocarbon oil, and covered with a 22 × 50 mm glass coverslip for live image analyses. Neurons expressing GFP were visualized on either a Nikon C1 Plus confocal microscope or a Zeiss LSM 780. For the Nikon C1 Plus microscope, images were collected as z-stacks using a 20X oil immersion lens at a step size of 2.5 µm and 1024 × 1024 resolution. For LSM 780, images were collected as z-stacks using a 20× dry lens at a step size of 2 µm and 1024 × 1024 resolution. The images were then converted to maximum intensity projections and were processed as previously described using ImageJ [19,26,31]. Quantitative morphometric data such as total dendritic length and the number of branches were extracted and compiled using a custom Python script, and the data output was imported into Microsoft Excel. Morphometric data were analyzed in Microsoft Excel and statistical tests were performed and graphs were plotted using GraphPad Prism 9.

For the Bdwf protein interactor cytoskeletal screen, *UAS-GMA;GAL4^477^,UASmCherry::Jupiter* [26] virgin flies were outcrossed to either *UAS-bdwf-IR, UAS-ct-IR* (B33967)*, UAS-RpS24-IR* (v104676), *UAS-RpL4-IR* (v101346), *UAS-RpL22-IR* (v104506), *UAS-RpL31-IR* (v104467) transgenic male flies or *Oregon R* males. Images were acquired from wandering third instar larvae on a Zeiss LSM780 confocal microscope using a 20× dry lens objective at a step size of 1.5 µm and 1024 × 1024 resolution. To analyze ribosome localization, *UAS-RpL10Ab::GFP; GAL4^221^,mCD8::RFP*, or *UAS-RPL10Ab::GFP; GAL4^ppk1.9^,UAS-mCD8::RFP* virgin female flies were crossed to *Oregon R* (control) or *UAS-bdwf-IR*. For ribosome localization, images were acquired at 63× and a step size of 1 µm and 1024 × 1024 resolution.

MARCM analyses were performed as previously described [19,26,33]. Briefly, *w, ct^c145^, FRT^19A^/y^+^, ct^+^, Y* males were outcrossed to *y,w,tubP-GAL80,hsFLP,FRT^19A^; GAL4^109(2)80^,UAS-mCD8::GFP* virgin females. Late third-instar larvae (96–120 hr after egg lay) were then examined for the presence of GFP labeled *cut* mutant clones, followed by dissection, fixation, and staining as described below for immunohistochemistry.

### 4.6. Multichannel Reconstructions

Multichannel cytoskeletal reconstructions and quantitative analysis were performed using a previously described method [38]. The ribosome signal was quantified following a similar procedure. Two-channel neuronal image stacks with membrane and ribosomal expression were processed to create multi-signal reconstructions [67]. Dendritic compartment local expression was quantified as follows:RQ = F × ASI × D
where RQ is the local ribosomal quantity and F is the fraction of the dendritic compartment volume occupied by the ribosomal signal. ASI is the average signal intensity and D is the local dendrite thickness, which is approximated by measuring the local dendritic diameter from the reconstructions. The average ribosomal expression per micrometer for each individual neuron was then calculated by dividing total ribosomal quantification by total length. Average expression specifically at the branch points was also calculated for each neuron. Both overall and branch-level average expression levels for all neurons were normalized and expressed as a fraction of the maximal average expression found in all neurons. All reconstruction data have been uploaded to NeuroMorpho.Org [68] (accessed on 1 February 2023)) and will be released in the Cox archive upon publication.

### 4.7. Generation of Anti-Bdwf Antibodies

The Bdwf protein sequence was obtained from flybase.org (CG3995-PA) and analyzed using a Hopp and Woods Antigenicity Plot to predict a high surface probability of putative Bdwf antigens that lie outside of the conserved Zinc-BED domain and exhibit protein sequence specificity unique to Bdwf protein. Based on these analyses, a distinct 15 amino acid peptide sequence (EFDVDEVEDVVPEED) in the middle of exon 2 was conjugated to KLH to improve the antigenicity of the peptide sequence, followed by polyclonal antibody production in rabbits (Genscript, Inc., Piscataway, NJ, USA). The anti-Bdwf antibody was affinity purified against the antigenic peptide, and specificity was confirmed by immunohistochemistry analyses.

### 4.8. Immunohistochemistry

Dissection, staining, and mounting of wandering third-instar larvae were performed as previously described [19]. For immunohistochemistry (IHC), larvae were dissected in PBS, pinned on Sylgard plates, and fixed in 4% paraformaldehyde for 20 minutes. Next, larvae were washed five times in 1× PBT (PBS + 0.3% Triton X-100). Cuticle filets were blocked for at least 30 minutes at room temperature in 5% normal donkey serum (Jackson Laboratories, West Grove, PA, USA), followed by incubation with respective primary antibodies. Antibody dilutions used were as follows: rabbit anti-Bdwf (1:200), mouse anti-CD8 (1:100) (Invitrogen), rabbit RpS6-(1:100) (Cell Signaling Technology, Danvers, MA, USA), and mouse anti-Cut (2B10; 1:50) (Developmental Studies Hybridoma Bank, Houston, TX, USA). Donkey anti-rabbit and donkey anti-mouse secondary antibodies were used at 1:200 (Jackson Immunoresearch, West Grove, PA, USA). Filets were incubated in glycerol for 5 minutes, followed by being directly mounted onto coverslips with either a drop of glycerol or an aqueous fluoromount. The slides were then imaged on a Nikon C1 Plus confocal microscope or a Zeiss LSM 780.

### 4.9. FUNCAT Analyses

FUNCAT analysis was performed as previously described [41,69,70]. Briefly, flies expressing *UAS-dMetRS^L624G^-3xmyc* under the control of *GAL4^ppk1.9^* were crossed to either *UAS-Luc-IR* (control) or *UAS-bdwf-IR* and grown in blue food containing 4 mM azidonorleucine (ANL) (Torcis, Cat No. 6585) and incubated at 29 °C. Wandering third instar larvae with visible blue dye in the guts were selected and dissected and fixed at 4% paraformaldehyde for 20 minutes. Larval fillets were washed with 1X PBT three times, followed by three washes with 1× PBS (pH 7.4) for 15 minutes each. Following washes, the fillets were treated with 1× PBS (pH 7.4) solution containing triazole ligand (200 mM TBTA, Sigma-Aldrich 1:1000 dilution, St. Louis, MI, USA), TAMRA-alkyne dye (Invitrogen, 1 mM, 1:5000 dilution), tris-(2-carboxyethyl) phosphine (400 mM, Sigma-Aldrich 1:1000 dilution), and CuSO_4_ (200 mM, Sigma Aldrich, 1:1000 dilution) and incubated overnight at 4 °C with gentle agitation. The fillets were then washed with 1× PBST (1× PBS pH 7.4 + 1% *v*/*v* Tween-20) the following day for 15 minutes each and then incubated with Alexa-Fluor goat anti-horseradish peroxidase (HRP) 488 (1:200) overnight at 4 °C. Following immunostaining, the fillets were washed in 1× PBT twice and once in 1× PBS (pH 7.4), mounted in an aqueous fluoromount, and imaged on a Zeiss LSM 780 at 63× objective.

### 4.10. Mass Spectrometry Analysis

Larval homogenates were prepared as described by Iyer et al. (2009). Age-matched third-instar larvae expressing *UAS-bdwf-FLAG-HA* driven by the ubiquitous *GAL4^Act5C^* driver were homogenized. Cell pellets were lysed in non-denaturing lysis buffer with protease inhibitors (50 mM Tris/HCl pH 7.5, 1 mM EGTA, 1 mM EDTA, 0.5 or 1% (*v*/*v*) NP-40, 1 mM sodium orthovanadate, 50 mM NaF, 5 mM sodium pyrophosphate, 0.27 M sucrose, 10 mM sodium 2-glycerophosphate, 0.2 mM phenylmethylsulphonyl fluoride, 1 mM benzamidine, plus 100 mM iodoacetamide added fresh before lysis, and pepstatin/aprotinin to inhibit proteases). Cell lysates were clarified by centrifugation at 14,000× *g* for 15 min at 4 °C. The protein concentration was determined after collecting the supernatant. Overnight, cell lysates were incubated with an anti-HA tag antibody. Bead-only or IgG-only controls were performed in parallel to control for specificity (Appendix A). To capture the protein of interest, 1 mg of cell extract protein was used and incubated for 3 h overnight at 4 °C with the affinity resin. The beads are then washed three times with 1 ml of lysis buffer containing 500 mM NaCl and once with 0.5 mL of 10 mM Tris/HCl pH 8.0.

The soluble protein samples from individual samples were dried with SpeedVac, reconstituted in 8 M urea, reduced by 10 mM DTT for 30 min, alkylated by 50 mM iodoacetamide for 30 min, and digested by trypsin at 37 °C overnight. Tryptic peptides were further purified by Zip-Tip (Millipore, Burlington, MA, USA) and analyzed by LC-MS/MS using a linear ion-trap mass spectrometer (LTQ, Orbitrap). After sample injection, the column was washed for 5 min with mobile phase A (0.4% acetic acid) and peptides eluted using a linear gradient of 0% mobile phase B (0.4% acetic acid, 80% acetonitrile) to 50% mobile phase B in 30 min at 250 nL/min, then to 100% mobile phase B for an additional 5 min. The LTQ mass spectrometer was operated in a data-dependent mode. After each full MS scan, five MS/MS scans were performed, during which the five most abundant molecular ions were dynamically selected for collision-induced dissociation with a normalized collision energy of 35%. Tandem mass spectra were collected by Xcalibur 2.0.2 and searched against the NCBI *Drosophila* protein database using SEQUEST (Bioworks 3.3.1 software from ThermoFisher, Waltham, MA, USA) using tryptic cleavage constraints. The mass tolerance for precursor ions was 5 ppm, and the mass tolerance for fragment ions was 0.25 Da. SEQUEST filter criteria were: Xcorr vs. charge 1.9, 2.2, and 3.5 for 1+, 2+, and 3+ ions; maximum probability of randomized identification of a peptide < 0.01. Protein identifications and the number of identifying spectra (peptide hits) for each sample were exported using a confidence level of >99% for protein identification, with peptides from a given protein identified at least in two independent samples (Appendix A). Functional enrichment analysis was performed using DAVID [71,72] to identify statistically overrepresented functional gene classes. For these analyses, the protein interactors of Bdwf were used as input, and all genes in the genome were selected as background. DAVID uses Fisher’s exact statistics to retain significant results calculated after redundancy in the original gene lists is removed (Appendix A). The EASE Score *p*-value ranges from 0 to 1, where a value of 0 represents perfect enrichment. *p*-values equal to or smaller than 0.05 were considered strongly enriched in the annotation categories.

## Figures and Tables

**Figure 1 ijms-24-06344-f001:**
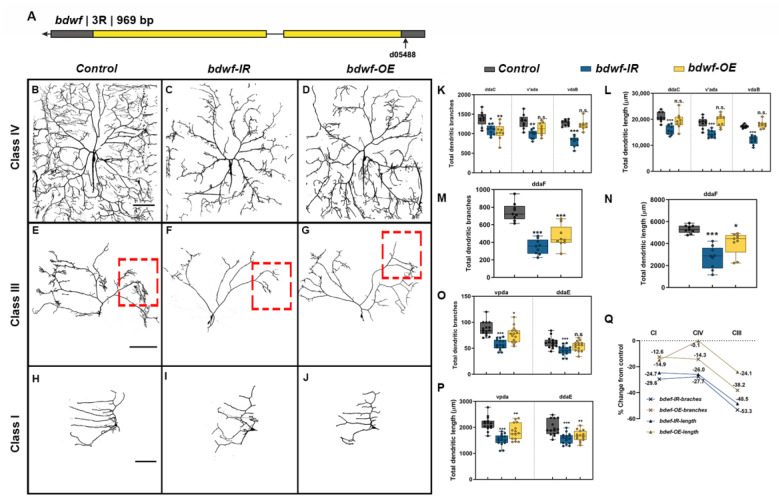
***Optimal levels of bdwf are essential for proper dendrite morphogenesis***. (**A**) Schematic diagram showing the *bdwf* gene and the location of the mutant allele used in the study. LOF and GOF result in strong and penetrant dendritic growth and branching defects in CIV (**B**–**D**), CIII (**E**–**G**), and CI md neurons (**H**–**J**). Scale bar, 50 µm. (**K**–**P**) Quantification of the total dendritic branches and total dendritic length are shown for CIV md neuron subtypes (ddaC, v’ada, and vdaB) (**K**,**L**), CIII md neuron subtype (ddaF) (**M**,**N**) and CI md neuron subtypes (ddaE, vpda) (**O**,**P**). * *p* ≤ 0.05, ** *p* ≤ 0.01, *** *p* ≤ 0.001, n.s. = not significant (One-way ANOVA with Dunnett’s multiple comparison test). Number of samples, *n* = 14–15 for CI, *n* = 9 for CIII and *n* = 8–9 for CIV. (**Q**) Total dendritic length and total dendritic branches for each condition are represented as percentage change from control. The dashed red boxes depicted in (**E,F,G**) correspond to the region displayed in **Figure 2** (**A**,**B**,**C**, respectively). Genotypes: (**B**–**D**,**K**,**L**,**Q**) *Gal4^477^, UAS*–*mCD8::GFP/+; +/+* or *Gal4^477^, UAS*–*mCD8::GFP/+; UAS*–*bdwf*–*IR/+* or *Gal4^477^, UAS*–*mCD8::GFP/+; UAS*–*bdwf*–*FLAG*–*HA/+*. (**E**–**G**,**M**,**N**,**Q**) *GAL4^ppk^,GAL80^ppk^,UAS*–*mCD8::GFP/+;+/+* or *GAL4^ppk^,GAL80^ppk^,UAS*–*mCD8::GFP/+;UAS*–*bdwf*–*IR/+* or *GAL4^ppk^,GAL80^ppk^,UAS*–*mCD8::GFP/+;UAS*–*bdwf*–*FLAG*–*HA/+*. (**H**–**J**,**O**–**Q**) *GAL4^221^,UAS*–*mCD8::GFP/+ or GAL4^221^,UAS*–*mCD8::GFP/UAS*–*bdwf*–*IR* or *GAL4^221^,UAS*–*mCD8::GFP/UAS*–*bdwf*–*FLAG*–*HA*.

**Figure 3 ijms-24-06344-f003:**
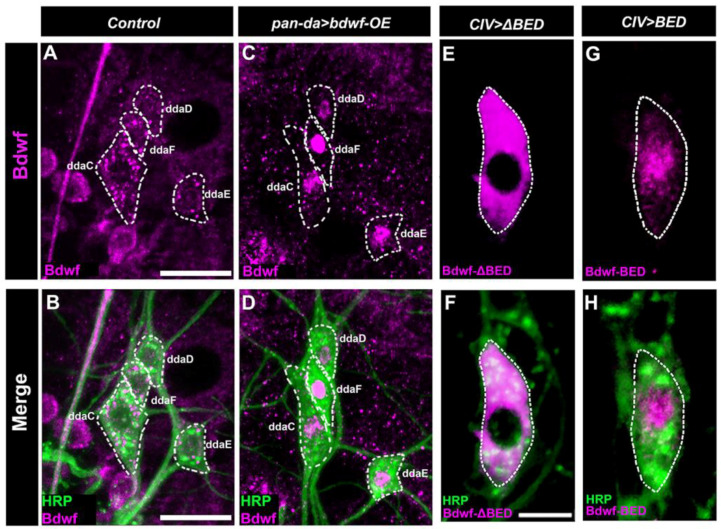
***Bdwf exhibits nucleo-cytoplasmic localization.*** (**A**–**D**) Representative confocal image of WT and Bdwf overexpressing dorsal cluster da neurons labeled with HRP (green) and anti-Bdwf antibody (magenta) to mark md neurons and Bdwf. md neuron cell bodies are highlighted by dotted white lines to represent Bdwf expression and localization in class I, III, and IV neurons. In wild-type, Bdwf shows weak nuclear localization and strong punctate cytoplasmic expression (**A**,**B**). Overexpression of *bdwf* transgene with the pan-da *GAL4^217^* driver produces a dramatic increase in nuclear localization (**C**,**D**) compared to wild-type while retaining punctate expression in the cytoplasm. Note that Bdwf is also expressed in puncta in surrounding epithelial and muscle tissue that is not labeled by HRP. Scale bar represents 20 µm. (**E**–**H**) BED domain is essential for Bdwf nuclearization. Representative confocal images of CIV md neurons expressing either *UAS-bdwf-BED* or *UAS-bdwf-ΔBED* using a class-IV specific driver. md neurons are immunolabeled with FITC-HRP (green) and anti-HA antibody (Magenta) to mark neurons and Bdwf mutant variants. Scale bar represents 5 µm. Genotypes: *GAL4^217^/+; +/+* (**A**,**B**), *GAL4^217^/+; UAS*–*bdwf*–*FLAG*–*HA/+* (**C**,**D**), *GAL4^477^/UAS*–*bdwf*–*ΔBED*–*myc; +/+* (**E**,**F**), *GAL4^477^/UAS*–*bdwf*–*BED*–*HA;+/+* (**G**,**H**).

**Figure 4 ijms-24-06344-f004:**
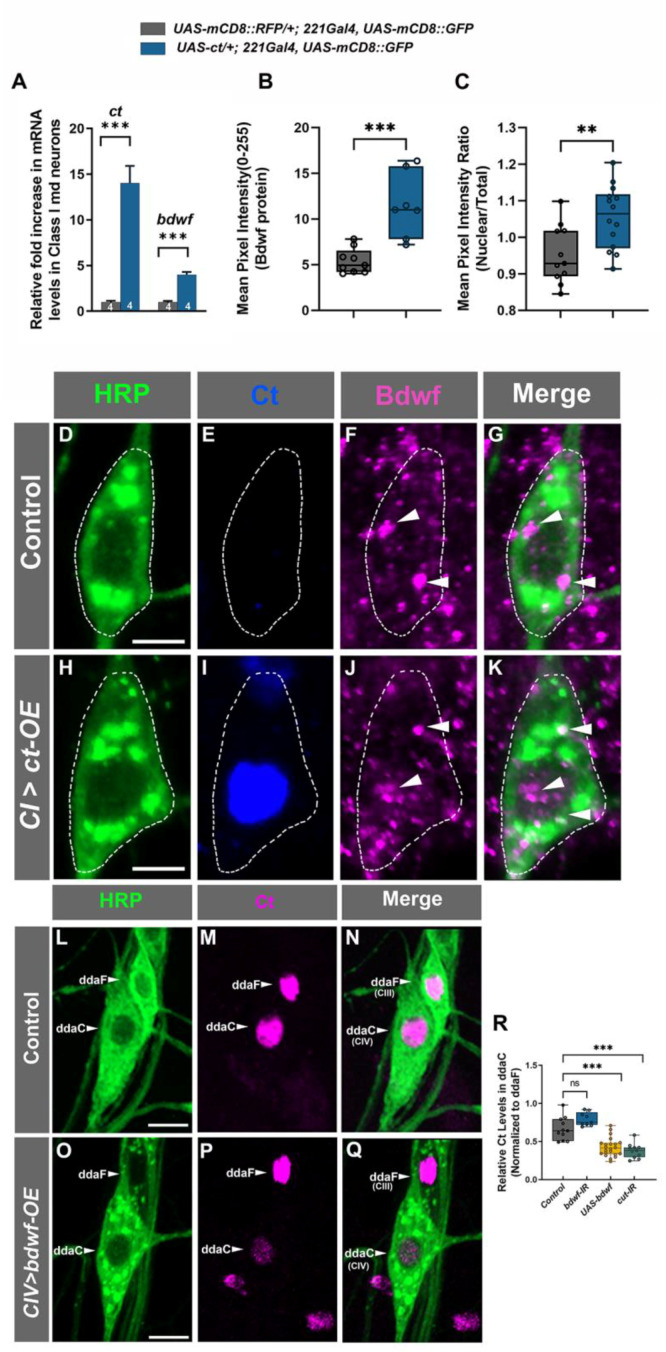
***Cut and Bdwf reciprocally regulate one another.*** (**A**–**K**) Cut positively regulates Bdwf expression. Immunohistochemical labeling of wild-type (**D**–**G**) and ectopically Cut expressing (**H**–**K**) ventral CI md neurons immunolabeled with HRP (green), Cut (blue) and Bdwf (magenta) antibodies. White dotted lines outline the CI md neuron cell bodies and white arrowheads point to punctate Bdwf expression in the nucleus and cytoplasm. (**L**–**Q**) Bdwf negatively regulates Cut expression. Compared to wild-type (**L**–**N**), Bdwf overexpression results in a strong and significant suppression of Cut immunostaining levels in CIV ddaC neurons (**O**–**Q**) immunolabeled with HRP (green) and Cut (magenta). Scale bar, 5 µm. (**A**) RT-PCR quantification of relative mRNA levels of Cut and Bdwf in wild-type and Cut-overexpressing class-I neurons. (**B**) Mean pixel intensity quantification Bdwf immunostaining in wild-type and Cut expressing CI vpda neurons. (**C**) Nuclear-to-cytoplasmic pixel intensity quantification of wild-type and Bdwf expressing CI vpda neurons showing increased nuclearization with Cut expression. (**R**) Mean pixel intensity of Cut in CIV ddaC neurons normalized to the CIII ddaF signal. Error bars represent +/− SEM. ** *p* ≤ 0.01, *** *p* ≤ 0.001, n.s. = not significant (Unpaired Student’s T-test or One-way ANOVA with Dunnett’s multiple comparison test). The number of samples (*n* value) for (**A**) is represented is indicated by the number inside the bar, for (**B**), *n* = 9 for WT and *n* = 9 for *ct-OE*, for (**C**) *n* = 11 for WT and 14 for *ct-OE*, for (**R**), *n* is between 9–22 per genotype. Genotypes: *UAS*–*cut; Gal4^221^, UAS*–*mCD8::GFP/+* or *+/+; Gal4^221^, UAS*–*mCD8::GFP/+* (**A**–**K**). *Gal4^477^, UAS*–*mCD8::GFP/+* or *Gal4^477^/UAS*–*mCD8::GFP/UAS*–*bdwf*–*FLAG*–*-HA* (**L**–**Q**), *Gal4^477^, UAS*–*mCD8::GFP/+* or *Gal4^477^/UAS*–*mCD8::GFP /UAS*–*bdwf*–*FLAG*–*HA* or *Gal4^477^/UAS*–*mCD8::GFP/+; UAS*–*bdwf*–*IR/+* or *Gal4^477^/UAS-mCD8::GFP /+*; *UAS-ct-IR/+* (**R**).

**Figure 5 ijms-24-06344-f005:**
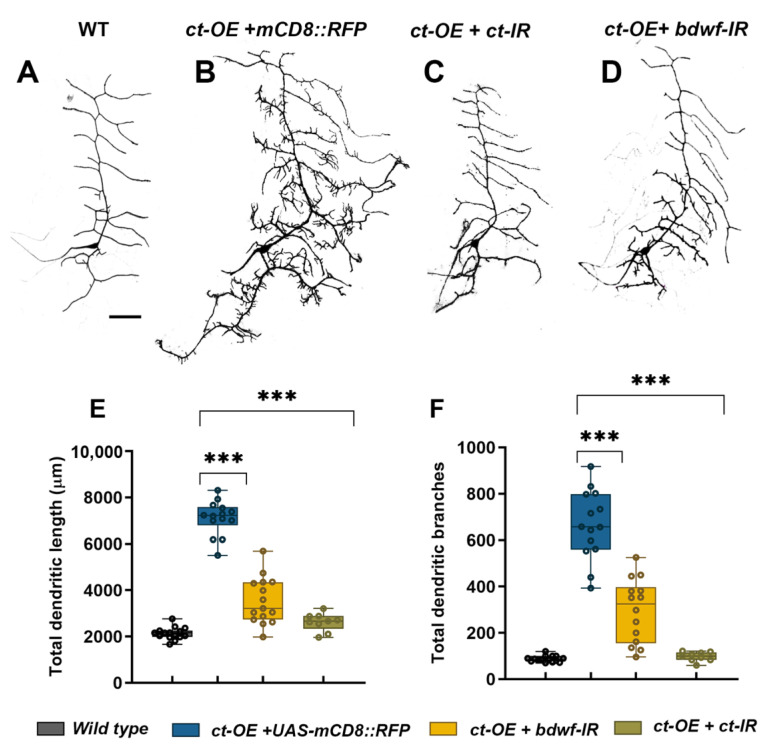
***Bdwf is required for Cut-induced dendritic development.*** (**A**–**D**) Suppression of Cut GOF phenotype in CI neuron by Bdwf LOF. Representative images of wild-type (**A**), Cut-overexpressing (**B**), Cut-overexpressing with Cut-LOF (**C**), and Cut overexpressing with Bdwf–LOF (**D**) CI vpda neurons. Scale bar, 50 µm. Quantification of TDL (**E**), and TDB (**F**), are shown. *** *p* ≤ 0.001 (one-way ANOVA with Dunnett’s test for multiple comparisons or Kruskal-Wallis with Dunn’s multiple correction test). *n* = 15 for WT, *n* = 14 for *ct-OE+UAS-mCD8::RFP*, *n* = 14 for *ct-OE+bdwf-IR*, *n* = 9 for *ct-OE+ct-IR*. Genotypes: *Gal4^221^, UAS*–*mCD8::GFP/+* (**A**,**E**,**F**), *UAS*–*ct/+; Gal4^221^, UAS*–*mCD8::GFP/+* (**B**,**E**,**F**), *UAS*–*ct/+; Gal4^221^, UAS*–*mCD8::GFP/UAS*–*ct*–*IR* (**C**,**E**,**F**), *UAS*–*ct/+; Gal4^221^, UAS*–*mCD8::GFP/UAS*–*bdwf*–*IR* (**D**–**F**).

**Figure 6 ijms-24-06344-f006:**
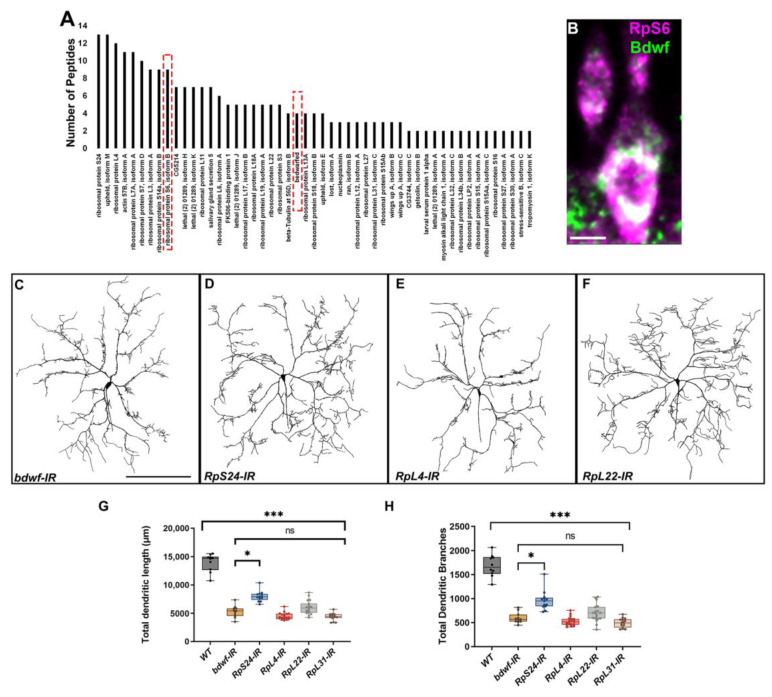
***Bdwf interacts and colocalizes with ribosomal proteins.*** (**A**) Bdwf-interacting proteins identified from MS analysis of affinity-purified Bdwf protein using our rabbit anti-Bdwf polyclonal antibody from homogenized third-instar larvae expressing *UAS-bdwf-FLAG-HA* protein driven by *GAL4^Act5C^* driver. Bdwf and ribosomal protein S6 are marked by a red dashed box. (**B**) Bdwf co-localizes with the 40 S ribosomal protein S6 to a high degree in md neurons. Knockdown of ribosomal proteins leads to a significant reduction in the TDL (**G**) and TDB (**H**) compared to the control. Except for *RpS24-IR*, knockdown of all the ribosomal proteins phenocopied *bdwf* knockdown with a high degree of penetrance. Representative images of *bdwf-IR*, *RpS24-IR*, *RpL4-IR*, and *RpL22-IR* (**C**–**F**). Scale bar, 10 µm (**B**), 200 µm (**C–F**). (**G**,**H**) Quantification of TDL and TDB in CIV ddaC neurons. * *p* ≤ 0.05, *** *p* ≤ 0.01, n.s. = not significant. *n* = 8 for WT, *n* = 11 for *bdwf-IR*, *n* = 13 for *RpS24-IR*, *n* = 19 for *RpL4-IR*, *n* = 19 for *RpL22-IR*, *n* = 15 for *RpL31-IR*. Genotypes: *GAL4^477^,UAS*–*mCD8::GFP/+; GAL4^ppk1.9^, UAS*–*mCD8::GFP/+*, or *GAL4^477^, UAS*–*-mCD8::GFP/+;GAL4^ppk1.9^, UAS*–*mCD8::GFP/UAS*–*bdwf*–*IR*, or *GAL4^477^,UAS*–*-mCD8::GFP/UAS*–*RpS24*–*IR;GAL4^ppk1.9^, UAS*–*mCD8::GFP/+*, or *GAL4^477^,UAS*–*mCD8::GFP/UAS*–*RpL4*–*IR;GAL4^ppk1.9^,UAS*–*mCD8::GFP/+*, *GAL4^477^,UAS*–*mCD8::GFP/+;GAL4^ppk1.9^,UAS*–*mCD8::GFP/UAS*–*RpL22*–*IR*, or *GAL4^477^,UAS*–*mCD8::GFP/UAS*–*RpL31*–*IR;GAL4^ppk1.9^,UAS*–*mCD8::GFP/+*.

**Figure 7 ijms-24-06344-f007:**
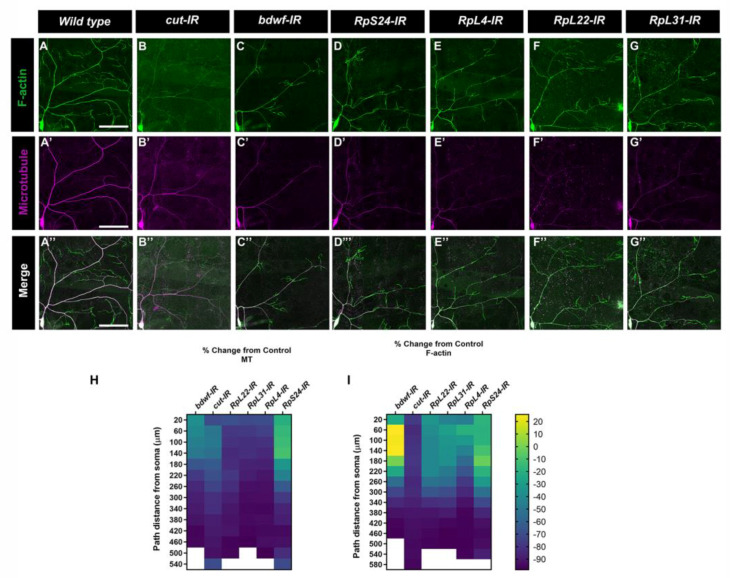
***Bdwf and Cut interact with ribosomal proteins to modulate the cytoskeletal architecture.*** (**A**–**G”**) Representative images of CIV ddaC neurons labeled with CIV-specific *UAS-GMA* (F-actin) and *UAS-mCherry::Jupiter* (MT). (**A**–**A”**) Control and (**B**–**G”**) gene-specific RNAi knockdown. (**H**,**I**). Heatmap showing the percent change from the control of MT signal (**H**) and F-actin signal (**I**) as a function of the path distance from the soma in knockdown conditions. *n* = 8 for *bdwf-IR*, *n* = 10 for *ct-IR*, *n* = 10 for *RpL22-IR*, *n* = 9 for *RpL31-IR*, *n* = 10 for *RpL4-IR*, *n* = 14 for *RpS24-IR*. Scale bar, 50 µm. Genotypes: *UAS*–*GMA;GAL4^477^,UAS*–*mCherry::JUPITER/+;+/+* (**A**–**A”**), or *UAS*–*GMA;GAL4^477^,UAS*–*mCherry::JUPITER/+; UAS*–*bdwf*–*IR/+* (**C**–**C”**,**H**,**I**), or *UAS*–*GMA;GAL4^477^,UAS*–*mCherry::JUPITER/+;UAS*–*ct*–*IR/+* (**B**–**B”**,**H**,**I**), or *UAS*–*GMA;GAL4^477^,UAS*–*mCherry::JUPITER/UAS*–*RpS24*–*IR;+/+* (**D**–**D”**,**H**,**I**), or *UAS*–*GMA;GAL4^477^,UAS*–*mCherry::JUPITER/UAS*–*RpL4*–*IR;+/+* (**E**–**E”**,**H**,**I**), or *UAS*–*-GMA;GAL4^477^,UAS*–*mCherry::JUPITER/+;UAS*–*RpL22*–*IR/+* (**F**–**F”**,**H**,**I**), or *UAS*–*GMA;GAL4^477^,UAS*–*mCherry::JUPITER/UAS*–*RpL31*–*IR;+/+* (**G**–**G”**,**H**,**I**).

**Figure 8 ijms-24-06344-f008:**
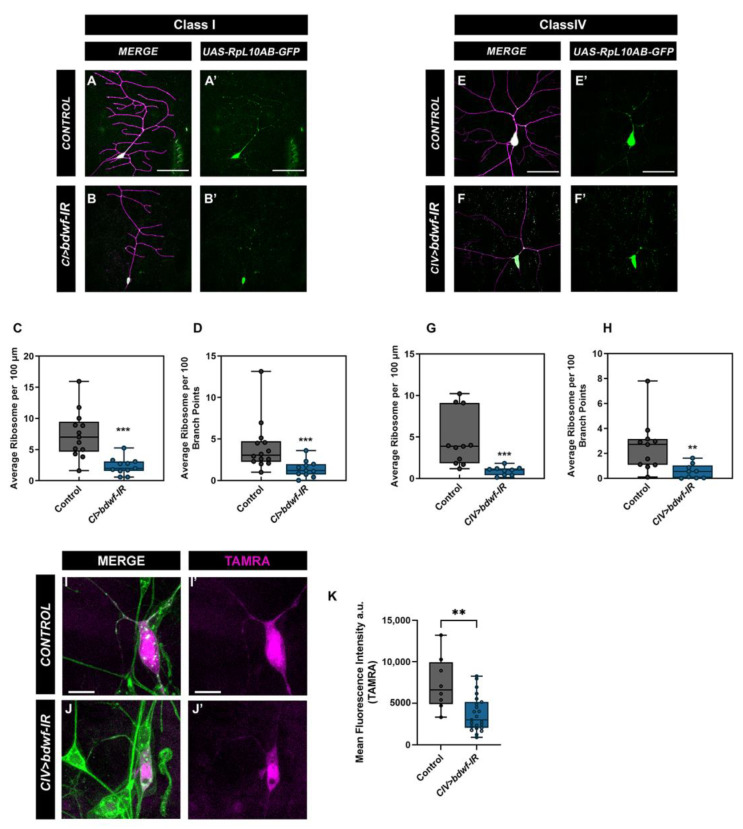
***Bdwf is required for ribosome trafficking along the dendrite and protein translation.*** Knockdown of *bdwf* shows that Bdwf is required for ribosome trafficking along the dendritic arbor in both CI and CIV md neurons (**A**–**H**). (**A**,**A’**,**B**,**B’**,**E**,**E’**,**F**,**F’**) Representative confocal image of WT and *bdwf-IR* CI and CIV md neurons, showing ribosome signal along the dendritic arbor, respectively. (**C**,**D**,**G**,**H**) Quantification of ribosome signal along the dendritic arbor normalized to length (**C**,**G**), or ribosome signal at branch points (**D**,**H**). Scale bar, 20 µm. Loss of Bdwf leads to disruption in protein translation (**I**,**J**). (**I**,**I’**,**J**,**J’**) Representative images of control and *bdwf-IR* expressing CIV neurons labeled with TAMRA (magenta) to show translated protein and dendrites labeled with HRP (green). Scale bar represents 10 µm. ** *p* < 0.01, *** *p* < 0.001(student’s T-test or Mann-Whitney test), *n*= 14 for CI control (**C**,**D**), *n* = 11 for CI *bdwf*–*IR* (**C**,**D**), *n* = 11 for CIV control (**G**,**H**), *n* = 9 for CIV *bdwf*–*IR* (**G**,**H**), *n* = 8 for CIV control and *n* = 23 for CIV *bdwf*–*IR* (**K**). Genotypes: *UAS*–*-RpL10Ab::GFP/+;GAL4^221^,UAS*–*mCD8::RFP/+* (**A**,**A’**), *UAS*–*RpL10Ab::GFP/+;GAL4^221^,UAS*–*mCD8::RFP/UAS*–*bdwf*–*IR* (**B**,**B’**), *UAS*–*RpL10Ab::GFP/+;GAL4^ppk^,UAS*–*mCD8::RFP/+* (**E**,**E’**), *UAS*–*RpL10Ab::GFP/+;GAL4^ppk^,UAS*–*mCD8::RFP/UAS*–*bdwf*–*IR* (**F**,**F’**), *UAS*–*dMetRS^L624G^*–*3xmyc/+;GAL4^ppk^/+* (**I**,**I’**), *UAS*–*dMetRS^L624G^*–*3xmyc/+;GAL4^ppk^/UAS*–*bdwf*–*IR* (**J**,**J’**).

**Figure 9 ijms-24-06344-f009:**
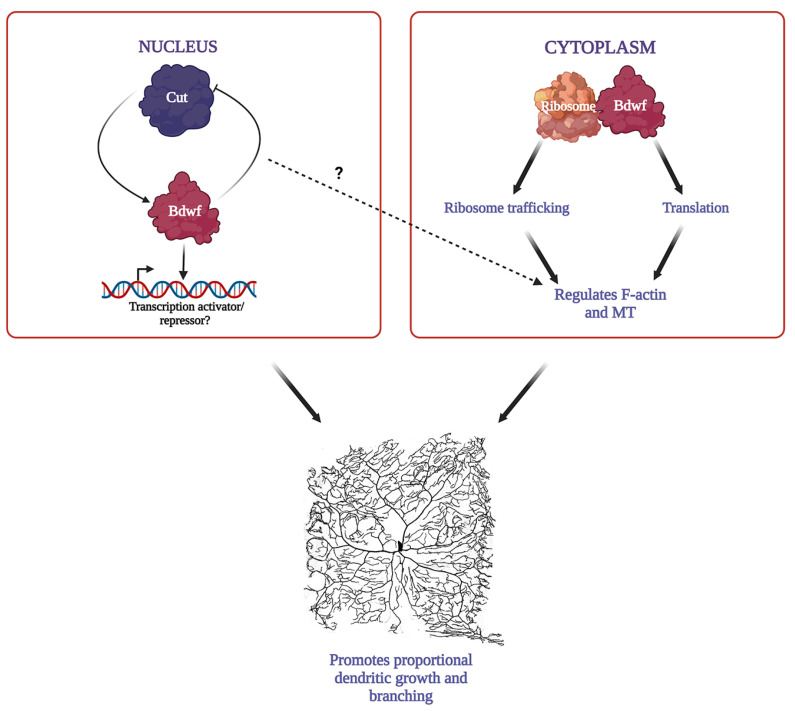
***Model for Bdwf function in dendrite morphogenesis.*** Cut and Bdwf have a reciprocal regulatory relationship whereby Cut positively regulates Bdwf expression, and overexpression of Bdwf negatively regulates Cut expression. Bdwf has putative DNA-binding activity. Bdwf interacts with ribosomal proteins to form a ribonucleoprotein complex, which potentially impacts ribosome trafficking and protein translation. Bdwf modulates cytoskeletal structure by promoting MT and F-actin-dependent dendritic growth and branching in Cut-positive neurons. Since both Cut and Bdwf regulate dendritic cytoskeletal components, it is currently unknown whether this occurs through a direct mechanism of nuclear Bdwf.

## Data Availability

The data presented in this study are available in the main article or Appendix A. All reconstruction data have been submitted to NeuroMorpho.Org (accessed on 1 February 2023) and will be available in the Cox archive upon publication.

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
