# Peer review of "The Zinc-BED Transcription Factor Bedwarfed Promotes Proportional Dendritic Growth and Branching through Transcriptional and Translational Regulation in Drosophila"

_ijms, 2023, doi:10.3390/ijms24076344_

Round 1

Reviewer 1 Report

In this manuscript, the authors addressed how cell-type specific dendritic patterning is regulated in terms of molecular mechanisms involved. Employing the Drosophila genetic toolkits, identified the functional role of a previously uncharacterized gene, CG3995, in multidendritic sensory neurons dendritic development. The authors show that the CG3995 gene, named bedwarfed, encodes an evolutionarily conserved zinc-finger BED-type protein that regulates dendrite growth and branching via two distinct mechanisms.  

The manuscript is written clearly and the experimental data supports the conclusion. Minor suggestions:

a. For easy readability, the authors should consider presenting the graphs in better color schemes and clear labeling.

b. Check the manuscript for typographical errors. For instance, GDP should be GFP. 

c. Include a brief note of neuronal subtypes -ddaC, v'ada, ddaF, vpda, etc

Author Response

We thank Reviewer 1 for their comments and include our response here:

  1. For easy readability, the authors should consider presenting the graphs in better color schemes and clear labeling.

We corrected an issue we found in Figure 1 and modified Figure S1 to have the same color scheme as other graphs.  In examining each figure, we found that the labeling was clear and the color schemes selected were to conform to best practices for color blind individuals.  If the reviewer has a specific recommendation for an alternate color scheme or specific issues with clarity, we are happy to make those edits, though we did not find other issues that impair the readability of the figure content.

  1. Check the manuscript for typographical errors. For instance, GDP should be GFP. 

We have checked the manuscript and made corrections for typographical errors.

  1. Include a brief note of neuronal subtypes -ddaC, v'ada, ddaF, vpda, etc

To improve the clarity around neuronal subtypes, we have inserted additional text on the following line numbers: 147, 151, 864, 865.

Reviewer 2 Report

The manuscript by Bhattacharjee et al. investigates the role of the transcription factor Bedwarfed in dendritic architecture in subtypes of drosophila neurones; demonstrating a regulatory loop between Bedwarfed and Cut in the nucleus and a distinct role in the cytoplasm where the TF associates to ribosomal protein to regulate protein translation.

The study is extremely well designed and elegantly conducted. The results support the authors' conclusions and advance the general knowledge regarding dendritogenesis in subtypes of neurones.

I have no particular comments, except that the manuscript is slightly wordy and might benefit from being shortened a little, to make it more straight to the point.

Well done.

Author Response

We thank the reviewer for their positive feedback on the manuscript.  We have re-examined the manuscript and in the interest of clarity, find that there was limited opportunity to notably reduce the text and still maintain the desired clarity and specificity of the content. 

Reviewer 3 Report

Review: The Zinc-BED transcription factor Bedwarfed promotes proportional dendritic growth and branching through transcriptional and translational regulation in Drosophila

This is a comprehensive manuscript describing  the transcription factor, Bdwf. The authors have shown a nucleocytoplasmic localization, an effect on neuronal sensory endings (i.e. dendrites). The  LOF and GOF phenotypic analyses strongly indicate a role. The images and analysis of branching md neuron subclasses is interesting as this opens new doors as to why the differences, such as activity patterns or local feedback control from neighboring cells, since  adjacent epithelial and/or muscle cells were mentioned to also contain Bdwf.  The mass spectroscopy analysis on identifying over 70 proteins interactors of  ribosomal proteins is dauting to think about how to dissect all the potential interactions.

The authors tell the story in a nice manner of the anatomical changes (LOF and GOF), localization, cell specific expression, MS analysis and the ribosomal trafficking along with a proposed model.

They covered all the bases and presented the data is an understandable manner. The figures and graphs as well explained and detailed. Methods are detailed well enough for one to know each procedure and how they were conducted.

I have no suggestions for the authors, besides congratulating them on a nice study and a well presented manuscript.

It would be interesting if Bdwf might add in the differential morphology observed in motor neuron terminals on larval muscles, as one can then address if the muscle might be regulating a local difference in ribosomal trafficking and Bdwf regulation.

Author Response

We thank the Reviewer for their positive comments on the manuscript.  We agree that investigating non-neuronal roles of Bdwf, given expression in muscle and epithelial cells, would be of interest in future studies, including potentially non-cell autonomous functions of Bdwf in regulating neuronal ribosomal trafficking.